# Applications of teledentistry in a French inmate population: A one-year observational study

Camille Inquimbert[1,2], Ioan Balacianu[3], Nicolas Huyghe[2], Joao Pasdeloup[2], Paul Tramini[2,4], Fadi Meroueh[3], Sylvie Montal[2], Sompop Bencharit [5]*, Nicolas Giraudeau[2,6]*

1 Systemic Health Care, University of Lyon 1, Lyon, France, 2 Dental Department, University Hospital of Montpellier, Montpellier, France, 3 Prison Medical Department, University Hospital of Montpellier, Montpellier, France, 4 LBN, University of Montpellier, Montpellier, France, 5 Department of General Practice, School of Dentistry and Department of Biomedical Engineering, College of Engineering, Virginia Commonwealth University, Richmond, Virginia, United States of America, 6 CEPEL, UMR 5112, University of Montpellier, CNRS, Montpellier, France

* sbencharit@vcu.edu (SB); nicolas.giraudeau@umontpellier.fr (NG)

## Abstract

Teledentistry oral examination protocol was evaluated for one year at the Villeneuve-lès-Maguelone Correctional Facility. The aim of the study was to simplify the obligatory dental consultation protocol at the entrance visit for new detainees. 1051 detainees were enrolled and 651 of them (58.9%) accepted an oral examination by teledentistry throughout the entire year of 2018. Only 1 inmate did not need treatment and 88.06% of those who have been examined had at least one untreated cavitated carious lesion. Forty-four percent of people who received a teledentistry check-up were referred to a dentist with a dental emergency. The use of teledentistry at the entry visit in a detention facility may facilitate the oral health screening without wasting the dentist's time, and may allow an optimization of the inmate's oral healthcare.

## Introduction

In France, all new inmates entering a prison have the right of having an entry medical check-up [1]. This consists of a visit with a physician, an interview with a psychologist, an interview with a nurse practitioner, and an oral check-up performed by a dentist. Within each prison health unit, a medical team is not always available to carry out appropriate entry medical checkups in one visit. Therefore, the oral and dental examination is only carried out in 52% of cases and very often by a health professional who is not a dentist [2]. This calls into question the quality of this entry check-up since only dentists are authorized and trained to make a dental assessment and proper dental diagnosis. Indeed, in France, the profession of second-tier dental professionals such as dental hygienists in the US does not exist. There is a shortage of dentists to carry out entry medical check-ups. The Medical Department of the Villeneuve- lès-Maguelone Arrest House (VLM), managed by the University Hospital Center (CHU) of

**Data Availability Statement:** All relevant data are within the manuscript and its supporting information files.

**Funding:** The operation of the project was funded by the Ministry of Health (FR) (DGS/SP1/N˚D.17.13667) to the Medical department of Villeneuve-lès-Maguelone's prison (University Hospital of Montpellier) and/or Dr Fadi Meroueh (as chief of the medical department). The funder has no role in the data collection, analysis, or the manuscript preparation or submission.

**Competing interests:** The authors have declared that no competing interests exist.

Montpellier therefore set up an oral telemedicine activity to fill in this gap. Initially, a trial experiment was carried out from 2015 to 2017 with funding from the Regional Health Agency (ARS) [3].

Telemedicine has been regulated in France since the telemedicine decree published in 2010 [4]. Since then telemedicine has been deployed throughout the country in order to fight against the demographic inequality of public healthcare. Teledentistry in France was born in 2014 with the e-DENT project led by the Montpellier University Hospital, which was first established to host dependent elderly people and people with disabilities [5]. In order to ensure the same quality of diagnosis by telemedicine as in face-to- face situations, an intraoral camera using fluorescence was chosen: the Soprocare® (Actéon Group, Mérignac, France). The study carried out by the Montpellier CHU showed a great similarity between the two techniques and validated the use of Soprocare® in the context of a telemedicine activity [6].

The inmate's oral health environment was compared to those with low socioeconomic status. There is a real need for an improvement of the existing oral health care system. The oral health of detainees is usually very poor characterized by a high incidence of dental caries, periodontal disease and missing teeth as shown by various studies in France and around the world [7–9]. This poor oral health may not only lead to general health problems, but also have a negative impact on the inmate's integration into his/her new environment in the prison as well as for his/her long-term reintegration into society and in the search for work after the prisoner is released.

The General Directorate of Health of the Ministry of Health launched a call for projects in 2017 to finance experiments aiming to improve oral dental consultations during the entrance visit. The Montpellier CHU responded and was selected for funding. The objective of the project was to offer all new entrants an oral check-up by teledentistry. The granted funding reached €55,353 for the 2018 year. As part of this study, we therefore analyzed the data collected during this project. The main objective was to analyze the implementation outcomes of oral teledentistry in establishing a baseline oral health of new inmates in the VLM Health Unit during the 2018 year. The secondary objectives were to evaluate the caries experience and the urgency level, and to collect information about the patient's primary care dentists.

## Materials and methods

The study protocol was approved by the General Directorate of Health of the Ministry of Health. This was a cross-sectional observational study for a period of one year, from January 3, 2018 to December 31, 2018 conducting at the Sanitary Unit of the VLM remand center, Emergency Department of the Montpellier University Hospital.

All new incomers were enrolled for the year of 2018 with the following inclusion and exclusion criteria (Table 1)

The oral examination through teledentistry activity took place in a dedicated room with the following set up:

• an armchair so that the patient can sit down and lie down slightly,

**Table 1. Inclusion and exclusion criteria.**

| Inclusion criteria: | Exclusion criteria: |
|---|---|
| Be a new member of the Villeneuve-lès- Maguelone remand center | Be in a mental state that does not allow the patient to understand the operator's instructions |
| Consent to benefit from an oral examination as part of the initial medical examination.* | Refuse oral examination via teledentistry |

*The study included all new inmates who accepted the oral examination through teledentistry.

- a stool with wheels so that the caregiver can sit facing the patient,

- a laptop computer on which is installed specific telemedicine software for collecting general information and videos,

- a pedal connected to the computer by a USB cable to start and stop the video captions,

- a Soprocare® intraoral camera connected to the computer by a USB cable.

A caregiver, a prison staff who was trained to accept the new inmate and to operate teledentistry processes, was equipped with examination gloves and a mask at the examination site with the new inmate. The caregiver was trained by the prison's dentist to do a virtual oral recording using Soprocare®, an intraoral camera using fluorescence light along with a video camera, and a computer.

The oral examination through teledentistry activity was taking place asynchronously between the caregiver and the remote dentist. The remote dentist received data through an internet connection at the end of the day. The dentist did not have to wait behind his computer all the time but could perform other tasks until the caregiver sent the files.

All new arrival inmates were offered an oral check-up by teledentistry by the caregiver. The inmates, who accepted the oral examination, were then accompanied to the room dedicated to this activity. An informed consent was obtained after the detailed explanation of the oral examination through teledentistry. The inmates were ensured that the examination data would be transferred safely to a secured server to allow analysis by a dentist from the public health service of the CHU of Montpellier. After the consent process the caregiver asked a number of following questions:

- Do you have a "primary care" dentist?

- If so, can you give its name or the type of structure (mutual center, CHU, other prison)

- What is your last dental visit? (1 month, 6 months, 1 year, 5 years, 10 years and more)

Then, the caregiver completed an odontogram, including all missing teeth and other oral lesions, for each inmate on the software by identifying the missing teeth on a pre-filled dental diagram. Finally, a video of each dental quadrant was produced. The caregiver recorded all the surfaces of all the teeth with the 2 fluorescence modes (periodontolgy mode and cariology mode). Any comments from the patient or the caregiver can be written to the remote dentist through the software. Once all the information was collected, it was sent securely to the server. The remote dentist analyzed them and identified the teeth that needed to be treated, already treated, and absent. The Decayed, Missing, and Filled Teeth (DMFT) score was then obtained. The medical team then was informed about the inmate's oral-dental condition. An urgency score based on a validated classification [10]. was appraised (Table 2). The report was sent back through the system and integrated into the inmate's file. Depending on the emergency score, an appropriate referral and a dental appointment if needed was then made. Since none of the indices often followed a normal distribution, a non-parametric Mann-Whitney test was used to compare quantitative variables among different groups. Statistical analysis was performed using Stata v16.1 software (Statacorp, Texas, United States). Fig 1 demonstrates the process of subject recruitment and teledentistry examination.

## Results

The primary outcomes of this work were the teledentistry implementation in the new inmates, while the secondary outcomes were oral health baseline data of the inmates once arrived at the prison, including past dental care, DMFT, and urgent care scoring. In 2018, 1,051 men entered

**Table 2. Urgency scores [10].**

| Score | Level of urgency | Description |
|---|---|---|
| 0 | No Urgency | No need for current treatment |
| 1 | No Urgency | Tooth cleaning and scaling needed |
| 2 | Low Urgency | Restorations and crowns needed, but none of them require immediate attention (restricted to the most superficial dentine). Include any person needing prosthesis or crowns. |
| 3 | Advanced Urgency | Deep enough restorations and crowns need attention right away (within 7–14 days) to avoid pulpal involvement or infection. Include any child with five or more teeth needing restorations |
| 4 | High Urgency | Requires urgent care due to pain or infection. Include here any person in need of pulpal treatment or extraction. |

VLM's remand center. Over 60% of them, 619 (58.90%), agreed to have an oral examination by teledentistry. The oral health data recording was done primarily by a caregiver with a remote dentist's as needed consultations. Through teledentistry, the remote dentist, which was the dentist responsible for the entire prison, could continue to treat patients all day with no interruptions for new consultation. The record data per inmate was done on average of 14 minutes and with additional 8 minutes on average of analysis time. It is important to note the main reasons for inmates to refuse oral examination by teledentistry. This included refusal of any medical exam, refusal of dental care or teledentistry, limited mouth opening or gag reflex, and fear of dentists.

In terms of oral health status of new inmates, out of the examined 619 people, 52 were edentulous and had removable complete denture prostheses and therefore no telemedicine assessment was performed. The remaining 567 (58.9%) inmates received an oral examination

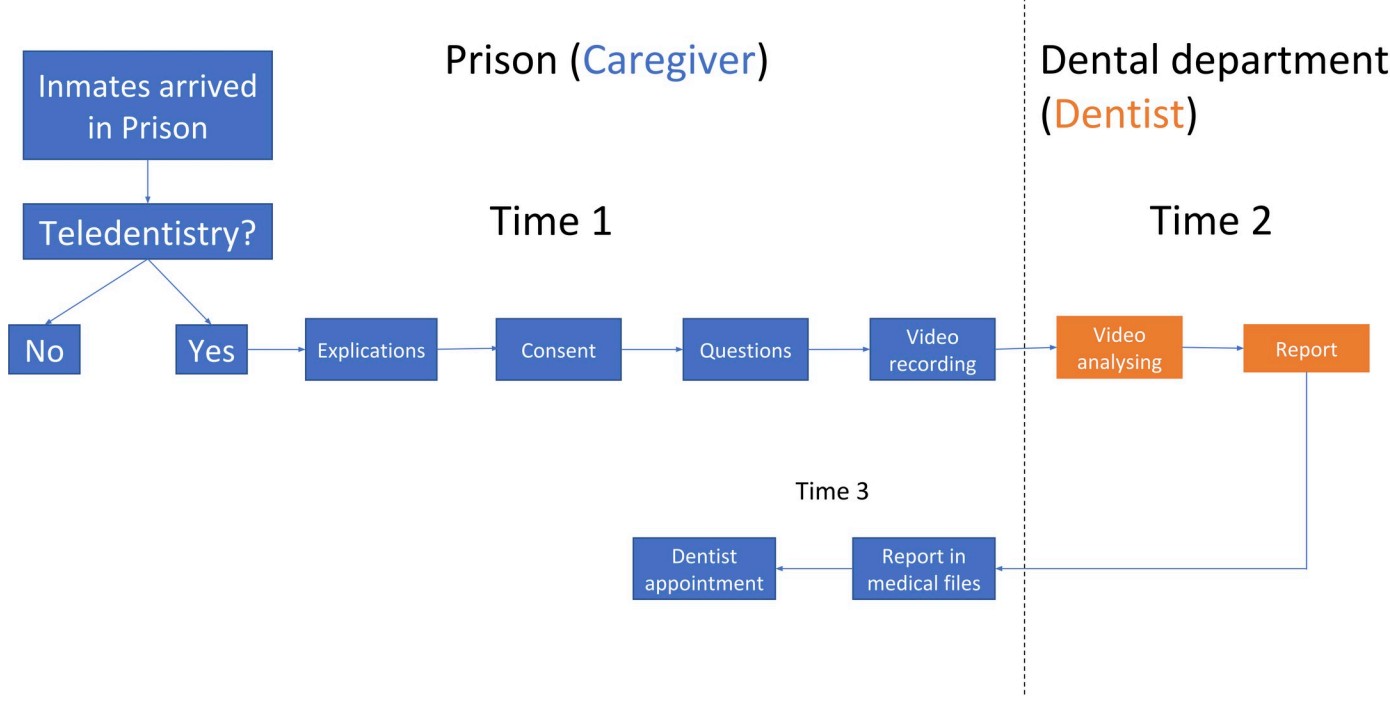

**Fig 1. Study workflow.**

**Table 3. Numbers of new inmates and oral teledentistry examinations.**

| Months | Jan | Feb | Mar | Apr | May | Jun | Jul | Aug | Sep | Oct | Nov | Dec | TOTAL |
|---|---|---|---|---|---|---|---|---|---|---|---|---|---|
| New comers | 76 | 89 | 75 | 101 | 87 | 100 | 89 | 71 | 102 | 78 | 103 | 80 | 1051 |
| Visit | 44 | 49 | 43 | 51 | 48 | 59 | 53 | 57 | 63 | 42 | 55 | 55 | 619 |
| Refusal | 32 | 40 | 32 | 50 | 39 | 41 | 36 | 14 | 39 | 34 | 48 | 25 | 430 |
| Prosthesis | 5 | 6 | 0 | 8 | 5 | 8 | 4 | 3 | 3 | 2 | 4 | 4 | 52 |
| % examined | 58% | 55% | 57% | 50% | 55% | 59% | 60% | 80% | 62% | 54% | 53% | 69% | 58.9% |

through teledentistry. The percentages of oral telemedicine oral examination range from the lowest 50% in April to the highest 80% in August (Table 3). The average age of new inmates is 31.65+/-10.69 years with the youngest of 16 and the oldest of 76 years old. Out of the 567 patients who underwent the telemedicine examination, 561 of them had sufficient accuracy oral teledentistry for further analysis for DMFT. Within this 561 inmates, we examined if having a primary care dentist and the last dental visit were factors influencing their oral health status. When asking a question: "Do you have a primary care dentist?", about half (51.68%) of the new inmates reported that they did not have a primary care dentist (Table 4). In terms of last dental visit, on average the inmate (out of 567 inmates) had seen a dentist 30.87+/- 36.78 months ago or just about two and a half years ago (Table 5).

The distribution of the DMFT score in this population of detainees is shown in Fig 2 and Table 6, with a mean value of 7.76 (+/- 5.31) and 9.27% caries-free. The mean components of DMFT were also assessed: D = 5.78 (+/- 4.64), M = 0.29 (+/- 0.82), F = 1.68 (+/- 2.93).

The mean number of cavitated lesions per inmate was 2.93 (3.11), and the mean number of enamel lesions was 0.87 (1.49). The mean number filled teeth with cavitated lesions was 0.74 (1.39). It is important to note that almost 90% of inmates have at least one tooth that needed to be treated (Table 6). The oral health status was generally very poor among prisoners. We identified only 9.27% (52) with a DMFT equal to 0 and only 12.12% (68) did not present any decay. Over 30% (34.40% or 193 inmates) showed a treated tooth with recovery of decay and only 46.17% (259) had at least one treated tooth. In terms of treatment urgency (Table 7), the average urgency score is less than 3 (referred to advanced urgency). However, over 40% of the inmates had an urgency score of 4 (high urgency) which required immediate treatment. About 12–31% of new inmates who received a teledentistry oral exam, subsequently had a dental appointment (Table 8).

To further examine if having a primary care dentist is a factor influencing the urgency score, DMFT, last dental visit, and extraction needed. Mann-Whitney U test was used to examine the differences between the inmate with and without a primary care dentist (Table 9). The mean urgency score was not significantly different when individuals had a regular primary care dentist or not (Table 9). New inmates with a primary care dentist had a higher DMFT score, reflective of better dental care, compared to the new inmates without a primary care

**Table 4. Numbers of inmates reported having a primary care dentist.**

| Primary care dentist | | | | |
|---|---|---|---|---|
| YES/No | Primary care dentist information | n | % | |
| NO | | 406 | 7.37% | 72.37% |
| YES | Doesn't know the name | 99 | 17.65% | 27.63% |
| | Knows the name | 50 | 8.91% | |
| | Others (CSERD, mutual, other US) | 6 | 1.07% | |
| | | 561 | | |

**Table 5. Reported last time seeing a dentist.**

| Last time seeing a dentist | n | % |
|---|---|---|
| 1 month | 65 | 11.46% |
| 6 months | 96 | 16.93% |
| 12 months | 225 | 39.68% |
| 60 months | 117 | 20.63% |
| 120 months | 64 | 11.29% |
| | 567 | 100% |

dentist (Table 9). The mean last appointment time was significantly different when individuals had a primary care dentist or not. New inmates with no primary care dentist also had not seen a dentist for a longer period of time compared to the ones with a primary care dentist (Table 9). However, the mean extraction needed was not significantly different when individuals had a primary care dentist or not (Table 9).

## Discussion

This first observational study is perhaps one of the first to examine the oral health of a large number of prison inmates during a one-year period using teledentistry. Our results suggest that teledentistry can be used as an innovative effective screening tool for oral health of prison inmates. Teledentistry therefore allowed us to screen and identify the oral conditions needing urgent interventions. The application of teledentistry as part of the initial medical exam can help establish oral health baseline and treatment needs for individual inmates. This would

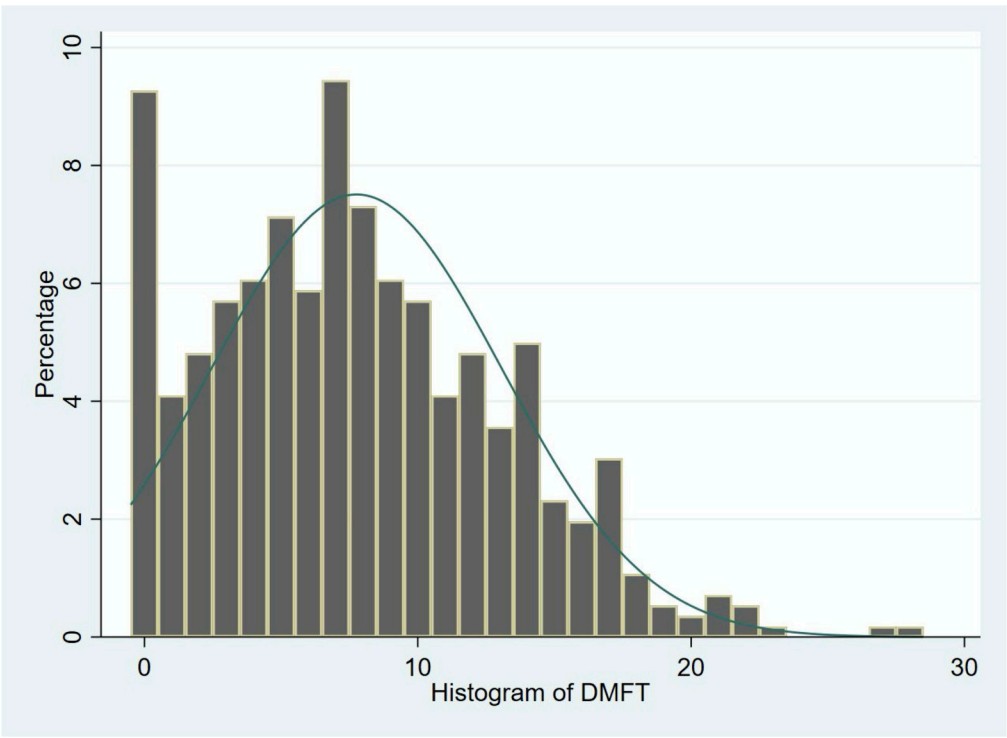

**Fig 2. Histogram of DMFT distributions.**

**Table 6. DMFT distribution\*.**

| Conditions | *n* | *%* |
|---|---|---|
| no decayed tooth | 68 | 12.12% |
| **at least one decayed teeth** | **493** | **87.88%** |
| no decayed filled tooth | 368 | 65.60% |
| **at least one decayed filled teeth** | **193** | **34.40%** |
| no extraction needed | 518 | 92.30% |
| **at least one extraction needed** | **43** | **7.70%** |
| no filled tooth | 302 | 53.83% |
| at least one filled tooth | 259 | 46.17% |
| no missing tooth | 461 | 82.17% |
| **at least one missing tooth** | **100** | **17.83%** |
| DMFT = 0 | 52 | 9.27% |
| DMFT>0 | 509 | 90.73% |

\*Diseased tooth conditions are represented in bold.

allow us to best integrate the new inmate into the captivity setting as well as establish long term care for them until they leave the detention facility.

Teledentistry is a functional and facilitating tool for the initial consultation on entry into detention and the organization of care. Thanks to the teledentistry activity set up for one year, the service was able to offer all new arrivals an oral examination. Nationally, initial oral exam check-up for new inmates is only offered for 52% of all inmates [11]. Previously, it had never been possible to find any organization in France that would allow this initiative. It is important to note that the use of teledentistry allows the diagnosis to be made by an oral health expert, a dentist, not by other health professionals who may not be competent in this discipline.

More importantly, teledentistry, in this case a virtual dental visit using a trained caregiver and a remote dentist, allows the dentist to concentrate on the patient care and treatment appointments that he has planned throughout the day and thus continue smoothly his activity without being impacted by the entrance examinations. Because the number of new inmate arrivals are often not known, without our screening teledentistry, a dentist may have to stand by doing nothing all day waiting for the inmate arrival. Indeed, the number of inmate arrivals can vary depending on the judicial activity and can be unlimited. Teledentistry makes it possible to avoid having to summon the detainee again after the arrival and also avoid unnecessary overcrowding in the dental department.

In our study, the images were reviewed by a practitioner in the dental department of the University Hospital of Montpellier, but the data analysis can also easily be done by the dentist

**Table 7. Distribution of urgency score\*.**

| | | |
|---|---|---|
| 0: no need for care | 1 | 0.18% |
| 1: need for a hygiene scaling | 59 | 10.41% |
| 2: low emergency | 231 | 40.74% |
| 3: advanced urgency | 14 | 2.47% |
| 4: high urgency | 234 | 41.27% |
| Undetermined | 28 | 4.94% |
| | 567 | 100.00% |

\*The average urgency score was 2.76 +/- 1.13.

**Table 8. Distribution of treatment after teledentistry.**

| Month | Jan | Feb | Mar | Apr | May | Jun | Jul | Aug | Sep | Oct | Nov | Dec | TOTAL |
|---|---|---|---|---|---|---|---|---|---|---|---|---|---|
| Appointment after teledentistry | 19 | 19 | 16 | 22 | 26 | 26 | 28 | 22 | 25 | 20 | 12 | 15 | 250 |
| % new comers | 25% | 21% | 21% | 22% | 30% | 26% | 31% | 31% | 25% | 26% | 12% | 19% | 24% |
| % visits | 43% | 39% | 37% | 43% | 54% | 44% | 53% | 39% | 40% | 48% | 22% | 27% | 40% |

working in the prison. The advantage of store-and-forward organization is that the prison dentist can analyze the data at the end of his day or when he has a moment in his working day (in the absence of a patient, or when he completed treatment prior to the plan) thus he optimizes his working time. Our protocol allows the same practitioner who analyzes the data to later provide care to the inmate if indicated afterwards. However, as we did in our study, agreement between the remote practitioner who analyzes and the one who will provide the care is very easy to set up.

It is important to point out the values of teledentistry in saving time and efforts for the remote dentist who had to take care of the entire prisoner population. The virtual visit with the help of a trained caregiver also allowed prioritization of care. On the other hand, this model can be implemented in a remote area, where patients may not be able to travel to see a dentist frequently. This would be a time and cost saving for patients or the public health system.

Telemedicine activity for prisoners has been implemented since 2013 [3]. However, the teledentistry implementation for new inmates has just been implemented in January 2018. Since then, this organization has continued and the number of refusals to the oral examination by telemedicine decreased. The use of oral teledentistry demanded a change in the detainees' management organization during the entrance visit; this sometimes took a little time, in order to change everyone's habits but today all people working in this department are delighted with the tool and process implementation. The doctor and / or nurses can learn about the inmates' oral health without disturbing the dentist since the oral telemedicine report is integrated into the patient's computerized medical record.

There is a very significant need for oral care in a population without a real care pathway beside the prison. Our study showed an average DMFT index of 7.76 (+/- 5.31) with a very large average number of decayed teeth at 5.78 (+/- 4.63), a number of missing teeth smaller than 1 (0.29 on average) and a low number of treated teeth. These figures are a little lower than in the latest studies identified in the literature. In Kosovo [8], the average DMFT was 8.44 in 2018 and was 10.6 in Sweden in 2018 [9]. The mean age of the male prison population is similar in the different studies. In our study, it reached 31.65 years of age. This relatively young age explains the low number of missing teeth. Indeed, only 17.83% (100) have at least one missing tooth. But 7.70% (43) need tooth extraction which may increase the number of missing teeth. No including total-denture-prosthesis wearers in this study also may lower the mean DMFT score.

**Table 9. Urgency score, DMFT, last dental appointment and extraction needed in subjects with and without a primary care dentist.**

| | n (%) | Urgency score | | DMFT | | Last dental appointment | | Extractions needed | |
|---|---|---|---|---|---|---|---|---|---|
| | | mean | SD | mean | SD | mean | SD | mean | SD |
| No primary care dentist | 406 (72.5%) | 2.76 | 1.15 | 7.37 | 5.26 | 36.71 | 39.04 | 0.14 | 0.65 |
| With a primary care dentist | 155 (27.5%) | 2.77 | 1.05 | 8.75 | 5.33 | 14.91 | 22.75 | 0.16 | 0.73 |
| Mann-Whitney test* | | 0.89 | | 0.005* | | 0.0001* | | 0.89 | |

*Statistical significant value ($\alpha = 0.05$).

The lack of care and the exit from the health system are also reflected in the identification of a treating dentist by the detainee. Indeed, only 27.5% (155) declared having a primary care dentist but only 8.91% (50) knew the name of their dentist. This may reflect a lack of involvement and motivation in their oral health care. This may explain the long average time of the last dentist appointment which was two and a half years. In a population with so many dental problems, this duration is far too long. The cross statistics show a correlation between the time spent since the last appointment and the identification of a practitioner. On the other hand, the study of the cross between the average DMFT on populations with a practitioner and those without any is interesting. The average DMFT among those who report having a practitioner is significantly greater when compared to those who do not report a treating practitioner. This can be explained by the fact that this population consults a lot on an emergency basis and that there is therefore very often a treatment that is carried out during the visit. Few members of this population have to go to the dentist for a check-up. The emergency score shows that 84.48% (479) of them needed care. The average score was 2.76 correlated with what inmates declared about having an outside the prison practitioner or not.

## Conclusions

The development of teledentistry is more and more under the spotlight and on the agenda as the World Health Organization has made it one of the modules of its mOralHealth program [12]. The prison population is a unique population which requires a special consideration and oral health care. The time spent in prison could be used to wisely "upgrade" the inmate's oral health and thus later facilitate their reintegration into society once the sentence has been served. The identification of oral pathologies upon entry into detention enables the prioritization and the organization of care. Teledentistry can be a real ally in achieving this goal.

Telemedicine is increasingly developing and the COVID-19 crisis that the world is experiencing is encouraging its use even further. This remote medical practice must be developed intelligently and in close collaboration and embark only the main objective of improving care and particularly of aiming to reduce inequalities in access to care. Telemedicine should not seek to replace a well-functioning organization that meets the needs of a population and / or a territory. Since there is a real need for oral health care in the inmate population, telemedicine appears to have a great value. It allows a systematization of the oral examination at the entry into custody by a competent professional. This would allow implementation of dental care based on the urgency as well as possible preventative care. It further optimizes the time of each actor for the direct benefit of the patient who is in great need of care.

## Supporting information

**S1 File.**
(XLSX)

## Author Contributions

**Conceptualization:** Fadi Meroueh, Sylvie Montal, Nicolas Giraudeau.

**Data curation:** Camille Inquimbert, Joao Pasdeloup, Paul Tramini, Nicolas Giraudeau.

**Formal analysis:** Paul Tramini.

**Funding acquisition:** Fadi Meroueh, Sylvie Montal, Nicolas Giraudeau.

**Investigation:** Camille Inquimbert, Ioan Balacianu, Nicolas Huyghe, Joao Pasdeloup, Nicolas Giraudeau.

**Methodology:** Camille Inquimbert, Ioan Balacianu, Nicolas Huyghe, Joao Pasdeloup, Paul Tramini, Fadi Meroueh, Sylvie Montal, Nicolas Giraudeau.

**Project administration:** Nicolas Giraudeau.

**Resources:** Fadi Meroueh, Sylvie Montal, Nicolas Giraudeau.

**Supervision:** Fadi Meroueh, Nicolas Giraudeau.

**Validation:** Paul Tramini, Sylvie Montal.

**Writing – original draft:** Camille Inquimbert, Nicolas Huyghe, Joao Pasdeloup, Paul Tramini, Sompop Bencharit, Nicolas Giraudeau.

**Writing – review & editing:** Sompop Bencharit, Nicolas Giraudeau.

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
