## [Decision Letter · Decision Letter 0]

29 Dec 2020

PONE-D-20-36871

Applications of Teledentistry in a French Inmate Population: A one-year observational study

PLOS ONE

Dear Dr. Bencharit,

Thank you for submitting your manuscript to PLOS ONE. After careful consideration, we feel that it has merit but does not fully meet PLOS ONE’s publication criteria as it currently stands. Therefore, we invite you to submit a revised version of the manuscript that addresses the points raised during the review process.

Please address concerns raised by both reviewers.

We look forward to receiving your revised manuscript.

Kind regards,

Abhishek Makkar, M.D.

Academic Editor

PLOS ONE

Journal Requirements:

2.In your Data Availability statement, you have not specified where the minimal data set underlying the results described in your manuscript can be found. PLOS defines a study's minimal data set as the underlying data used to reach the conclusions drawn in the manuscript and any additional data required to replicate the reported study findings in their entirety. All PLOS journals require that the minimal data set be made fully available. For more information about our data policy, please see http://journals.plos.org/plosone/s/data-availability.

3. Please ensure that you refer to Figure 2 in your text as, if accepted, production will need this reference to link the reader to the figure.

4. We note you have included a table to which you do not refer in the text of your manuscript. Please ensure that you refer to Table 5, 8, 9, 10 and 11 in your text; if accepted, production will need this reference to link the reader to the Table.

Reviewers' comments:

Reviewer's Responses to Questions

**Comments to the Author**

1. Is the manuscript technically sound, and do the data support the conclusions?

Reviewer #1: Partly

Reviewer #2: Partly

2. Has the statistical analysis been performed appropriately and rigorously? 

Reviewer #1: I Don't Know

Reviewer #2: No

3. Have the authors made all data underlying the findings in their manuscript fully available?

Reviewer #1: Yes

Reviewer #2: Yes

4. Is the manuscript presented in an intelligible fashion and written in standard English?

Reviewer #1: No

Reviewer #2: No

5. Review Comments to the Author

Reviewer #1: Applications of Teledentistry in a French Inmate Population: A one-year observational

Study

Objectives

The main objective was to analyze the implementation of oral telemedicine in the VLM Health Unit during the 2018 year.

The secondary objectives were to evaluate the caries experience and the urgency level, and to collect information about the patients’ practitioners.

Materials and Methods

• DMFT Score - The DMFT score was then obtained- Please describe what the DMFT score is as all the results were focused on this and there is no mention of the description of the score for a reader.

• Please explain the caregiver role in methods – is he/she a person with previous dental provider experience or a novice person who was taught how to perform oral examinations with telemedicine equipment

Results

Overall, the results section is incomplete and vague. Please summarize all the results in 1-2 paragraphs with corresponding references to the tables. There were lot of tables and figures with no legends. Multiple tables were used, which could be summarized into fewer tables.

• Conditions making the assessment difficult or impossible such severe gag reflect – sentence not clear

• Table 4 Reported last time seeing a dentist

-Please label the columns, it’s not clear especially what the data in third column meant

• Table 5 DMFT distribution

It would be ideal to mention the abnormalities to get a clear picture, Example below

Abnormalities n (%)

at least one decayed teeth 493 (87.88%)

at least one decayed filled teeth 138 (34.40%)

• Table 7 Distribution of treatment after teledentistry

Table 7 top 4 rows replicates the data which was already presented in table 2, please redo this table to minimize duplication

• Tables 8 through 11 and Figure 2

- All these tables basically compare NO PRACTITIONER AND REGULAR PRACTITIONER group differences. These all could be summarized in a single table rather than 4 different tables and figure 2. This allows the reader to better understand the message being conveyed.

-The number in NO PRACTITiONER and REGULAR PRACTITIONER groups was different in table 8 and Table 9 through 11

Discussion:

• There is duplication of results within the discussion section. Please avoid the duplication if possible.

• It would be ideal if the authors can summarize their key findings from the study at the beginning of the discussion section to set forth further discussion.

• Some of the the key findings from the study seem to be

o Observational study of oral health of large number of prison inmates

o Teledentistry used as an innovative effective screening tool for oral health of prison inmates

o Telemedicine allowed to screen and identify the oral conditions needing urgent interventions

• Once the key results are summarized, the authors can then discuss each key finding with supported literature in a concise manner

• The authors did not highlight any limitations of the study

Reviewer #2: Thank you for the valuable study, it is a great addition to the literature and provides insights about potential valuable use of Telehealth in both preventive and diagnostic dentistry.

It would be great to reorganize the paper. Some suggestions in the comment section.

Thank you

6. PLOS authors have the option to publish the peer review history of their article (what does this mean?). If published, this will include your full peer review and any attached files.

Reviewer #1: No

Reviewer #2: No

---

## [Author Response · Author response to Decision Letter 0]

20 Jan 2021

RESPONSES TO EDITOR AND REVIEWERS

The authors are grateful for the kind comments and thorough reviews from the editor and reviewers. The following list includes all comments, responses, and text modifications. 

RESPONSE: The manuscript has been revised per PLOS ONE’s style requirements.

2.In your Data Availability statement, you have not specified where the minimal data set underlying the results described in your manuscript can be found. PLOS defines a study's minimal data set as the underlying data used to reach the conclusions drawn in the manuscript and any additional data required to replicate the reported study findings in their entirety. All PLOS journals require that the minimal data set be made fully available. For more information about our data policy, please see http://journals.plos.org/plosone/s/data-availability.

RESPONSE: The de-identified data sheet in an excel format was added as supplementary information.

3. Please ensure that you refer to Figure 2 in your text as, if accepted, production will need this reference to link the reader to the figure.

RESPONSE: Figure 2 demonstrated redundant data that are displayed already in the Tables. We remove Figure 2 per Reviewer #1’s request. 

4. We note you have included a table to which you do not refer in the text of your manuscript. Please ensure that you refer to Table 5, 8, 9, 10 and 11 in your text; if accepted, production will need this reference to link the reader to the Table.

RESPONSE: Tables are now revised, consolidated and cited.

Reviewers' comments:

Reviewer's Responses to Questions

Comments to the Author

Reviewer #1: Applications of Teledentistry in a French Inmate Population: A one-year observational

Study

Objectives

The main objective was to analyze the implementation of oral telemedicine in the VLM Health Unit during the 2018 year.

The secondary objectives were to evaluate the caries experience and the urgency level, and to collect information about the patients’ practitioners.

Materials and Methods

• DMFT Score - The DMFT score was then obtained- Please describe what the DMFT score is as all the results were focused on this and there is no mention of the description of the score for a reader.

RESPONSE: We appreciate the comment.

TEXT CHANGE: DMFT definition as Decay, Missing, and Filled Tooth was added.

• Please explain the caregiver role in methods – is he/she a person with previous dental provider experience or a novice person who was taught how to perform oral examinations with telemedicine equipment

RESPONSE: We appreciate the comment. The caregiver role/training was added in the Methods. 

TEXT CHANGE: “The caregiver was trained by the prison’s dentist to do a virtual oral examination using Soprocare®, an intraoral camera along with a video camera, and a computer. The caregiver was also trained in simple oral diagnosis with a virtual consulted dentist.”

Results

Overall, the results section is incomplete and vague. Please summarize all the results in 1-2 paragraphs with corresponding references to the tables. There were lot of tables and figures with no legends. Multiple tables were used, which could be summarized into fewer tables.

RESPONSE: We appreciated the comments very much. The results section has been rewritten per your suggestion. The Tables are consolidated. Figure 2 was removed (redundant data). Legends are added. 

TEXT CHANGE: Please see detailed changes below.

“The primary outcomes of this work were the teledentistry implementation in the new inmates, while the secondary outcomes were oral health baseline data of the inmates once arrived at the prison, including past dental care, DMFT, and urgent care scoring. In 2018, 1,051 men entered VLM's remand center. Over 60% of them, 619 (58.90%), agreed to have an oral examination by teledentistry. The oral health data recording was done primarily by a caregiver with a remote dentist’s as needed consultations. Through teledentistry, the remote dentist, which was the dentist responsible for the entire prison, could continue to treat patients all day with non interruptions for new consultation. The record data per inmate was done on average of 14 minutes and with additional 8 minutes on average of analysis time. It is important to note the main reasons for inmates to refuse oral examination by teledentistry. This included refusal of all or part of the entry medical examination, having limited mouth opening or severe gag reflect that cannot tolerate the intraoral device, conditions making the assessment difficult or impossible such severe gag reflect, refusal of benefit from dental care, refusal of teledentistry, and finally fear of dentists.

In terms of oral health status of new inmates, out of the examined 619 people, 52 were edentulous and had removable complete denture prostheses and therefore no telemedicine assessment was performed. The remaining 567 (58.9%) inmates received an oral examination through teledentistry. The percentages of oral telemedicine oral examination range from the lowest 50% in April to the highest 80% in August (Table 3). The average age of new inmates is 31.65+/-10.69 years with the youngest of 16 and the oldest of 76 years old. Out of the 567 patients who underwent the telemedicine examination, 561 of them had sufficient accuracy oral teledentistry for further analysis for DMFT. Within this 561 inmates, we examined if having a primary care dentist and the last dental visit were factors influencing their oral health status. When asking a question: "Do you have a primary care dentist?”, about half (51.68%) of the new inmates reported that they did not have a primary care dentist (Table 4). In terms of last dental visit, on average the inmate (out of 567 inmates) had seen a dentist 30.87+/- 36.78 months ago or just about two and a half years ago (Table 5). 

The distribution of the DMFT score in this population of detainees is shown in Figure 2 and Table 6, with a mean value of 7.76 (+/- 5.31) and 9.27% caries-free. The mean components of DMFT were also assessed: D = 5.78 (+/- 4.64), M = 0.29 (+/- 0.82), F = 1.68 (+/- 2.93). 

The mean number of cavitated lesions per inmate was 2.93 (3.11), and the mean number of enamel lesions was 0.87 (1.49). The mean number filled teeth with cavitated lesions was 0.74 (1.39). It is important to note that almost 90% of inmates have at least one tooth that needed to be treated (Table 6). The oral health status was generally very poor among prisoners. We identified only 9.27% (52) with a DMFT equal to 0 and only 12.12% (68) did not present any decay. Over 30% (34.40% or 193 inmates) showed a treated tooth with recovery of decay and only 46.17% (259) had at least one treated tooth. In terms of treatment urgency (Table 7), the average urgency score is less than 3 (referred to advanced urgency). However, over 40% of the inmates had an urgency score of 4 (high urgency) which required immediate treatment. About 12-31% of new inmates who received a teledentistry oral exam, subsequently had a dental appointment (Table 8). 

To further examine if having a primary care dentist is a factor influencing the urgency score, DMFT, last dental visit, and extraction needed. Mann-Whitney U test was used to examine the differences between the inmate with and without a primary care dentist (Table 9). The mean urgency score was not significantly different when individuals had a regular primary care dentist or not (Table 9). New inmates with a primary care dentist had a higher DMFT score, reflective of better dental care, compared to the new inmates without a primary care dentist (Table 9). The mean last appointment time was significantly different when individuals had a primary care dentist or not. New inmates with no primary care dentist also had not seen a dentist for a longer period of time compared to the ones with a primary care dentist (Table 9). However, the mean extraction needed was not significantly different when individuals had a primary care dentist or not (Table 9).”

• Conditions making the assessment difficult or impossible such severe gag reflect – sentence not clear

RESPONSE: We agree.

TEXT CHANGE: The sentence is now read:

“Having limited mouth opening or severe gag reflect that cannot tolerate the intraoral device”

• Table 4 Reported last time seeing a dentist

-Please label the columns, it’s not clear especially what the data in third column meant

RESPONSE: We appreciate the comment.

TEXT CHANGE: Each column is now labelled.

• Table 5 DMFT distribution

It would be ideal to mention the abnormalities to get a clear picture, Example below

Abnormalities n (%)

at least one decayed teeth 493 (87.88%)

at least one decayed filled teeth 138 (34.40%)

RESPONSE: We appreciate the comment.

TEXT CHANGE: Diseased tooth conditions are now represented in bold with a footnote labelled.

• Table 7 Distribution of treatment after teledentistry

Table 7 top 4 rows replicates the data which was already presented in table 2, please redo this table to minimize duplication

RESPONSE: We appreciate the comment.

TEXT CHANGE: The top 5 rows were removed. 

• Tables 8 through 11 and Figure 2

- All these tables basically compare NO PRACTITIONER AND REGULAR PRACTITIONER group differences. These all could be summarized in a single table rather than 4 different tables and figure 2. This allows the reader to better understand the message being conveyed.

-The number in NO PRACTITiONER and REGULAR PRACTITIONER groups was different in table 8 and Table 9 through 11

RESPONSE: We thank the reviewer for this insight. 

TEXT CHANGE: Table 8 through Table 11 are consolidated into one table. Figure 2 was removed. The numbers in Table 8 were corrected.

Discussion:

• There is duplication of results within the discussion section. Please avoid the duplication if possible.

RESPONSE: We really appreciate the input.

TEXT CHANGE: All redundant results were removed from the Discussion to avoid duplication.

• It would be ideal if the authors can summarize their key findings from the study at the beginning of the discussion section to set forth further discussion.

• Some of the the key findings from the study seem to be

o Observational study of oral health of large number of prison inmates

o Teledentistry used as an innovative effective screening tool for oral health of prison inmates

o Telemedicine allowed to screen and identify the oral conditions needing urgent interventions

RESPONSE: We really appreciate this suggestion

TEXT CHANGE: The key finding paragraph was added at the beginning of the Discussion.

• Once the key results are summarized, the authors can then discuss each key finding with supported literature in a concise manner

RESPONSE: We appreciate this suggestion

TEXT CHANGE: The supportive literature was now discussed following the key finding paragraph.

• The authors did not highlight any limitations of the study

RESPONSE: We appreciate this suggestion

TEXT CHANGE: A limitation paragraph was added at the end of the Discussion section.

Reviewer #2: Thank you for the valuable study, it is a great addition to the literature and provides insights about potential valuable use of Telehealth in both preventive and diagnostic dentistry.

It would be great to reorganize the paper. Some suggestions in the comment section.

RESPONSE: The authors truly appreciate the guidance from the reviewer. We also thank the reviewer for the time and effort spending on reviewing the manuscript. We included all corrections and suggestions in the revised manuscript. Please see the manuscript with track changes.

---

## [Editor Report · Decision Letter 1]

5 Feb 2021

PONE-D-20-36871R1

Applications of Teledentistry in a French Inmate Population: A one-year observational study

PLOS ONE

Dear Dr. Bencharit,

Thank you for submitting your manuscript to PLOS ONE. After careful consideration, we feel that it has merit but does not fully meet PLOS ONE’s publication criteria as it currently stands. Therefore, we invite you to submit a revised version of the manuscript that addresses the points raised during the review process.

Thanks for taking time to address most of reviewer comments. Please address following before we can consider your manuscript for publication.

**Introduction**: Last line of Introduction should be revised to say Patient's Primary care dentist instead of Patient's Practioner to be consistent with terminology used in other sections of  manuscript.

**Results: **Please rephrase Line 10 of results, it is hard to read and needs grammatical correction. Sentence starting. This included refusal of ………….fear of dentists.

**Tables**: Please check formatting on table 3,6 and 8 so two or 3 digit  numbers displayed are in one line.

Please complete missing column lables in Tables 5 and 7. ( Currently missing n and % label)

**References: **Please translate references 1,2,4,5,7,11 to English.

Reference 10 is incomplete, please submit complete reference with online access date if its a webpage.

We look forward to receiving your revised manuscript.

Kind regards,

Abhishek Makkar, M.D.

Academic Editor

PLOS ONE

---

## [Author Response · Author response to Decision Letter 1]

9 Feb 2021

RESPONSES TO COMMENTS FROM EDITOR/REVIEWERS

Introduction: Last line of Introduction should be revised to say Patient's Primary care dentist instead of Patient's Practitioner to be consistent with terminology used in other sections of manuscript.

RESPONSE: Appreciate the comment.

TEXT CHANGE: The sentence is now read: “... patient’s primary care dentist.’

Results: Please rephrase Line 10 of results, it is hard to read and needs grammatical correction. Sentence starting. This included refusal of ………….fear of dentists.

RESPONSE: We appreciate the comment.

TEXT CHANGE: The sentence is now read: “This included refusal of any medical exam, refusal of dental care or teledentistry, limited mouth opening or gag reflex, and fear of dentists.”

Tables: Please check formatting on table 3,6 and 8 so two or 3 digit numbers displayed are in one line.

RESPONSE: We appreciate the comment.

TEXT CHANGE: Changes have been made.

Please complete missing column lables in Tables 5 and 7. ( Currently missing n and % label)

RESPONSE: We appreciate the comment.

TEXT CHANGE: Changes have been made.

References: Please translate references 1,2,4,5,7,11 to English.

RESPONSE: We appreciate the comment.

TEXT CHANGE: Changes have been made.

Reference 10 is incomplete, please submit complete reference with online access date if its a webpage.

RESPONSE: We appreciate the comment.

TEXT CHANGE: Changes have been made.

---

## [Editor Report · Decision Letter 2]

16 Feb 2021

Applications of Teledentistry in a French Inmate Population: A one-year observational study

PONE-D-20-36871R2

Dear Dr. Bencharit,

We’re pleased to inform you that your manuscript has been judged scientifically suitable for publication and will be formally accepted for publication once it meets all outstanding technical requirements.

Kind regards,

Abhishek Makkar, M.D.

Academic Editor

PLOS ONE
---

## [Editor Report · Acceptance letter]

29 Mar 2021

PONE-D-20-36871R2 

Applications of Teledentistry in a French Inmate Population: A one-year observational study 

Dear Dr. Bencharit:

I'm pleased to inform you that your manuscript has been deemed suitable for publication in PLOS ONE. Congratulations! Your manuscript is now with our production department. 

Kind regards, 

on behalf of

Dr. Abhishek Makkar 

Academic Editor

PLOS ONE